# Lead Disrupts Mitochondrial Morphology and Function through Induction of ER Stress in Model of Neurotoxicity

**DOI:** 10.3390/ijms231911435

**Published:** 2022-09-28

**Authors:** Jianbin Zhang, Peng Su, Chong Xue, Diya Wang, Fang Zhao, Xuefeng Shen, Wenjing Luo

**Affiliations:** Department of Occupational & Environmental Health and the Ministry of Education Key Lab of Hazard Assessment and Control in Special Operational Environment, School of Public Health, Fourth Military Medical University, 169 Changlexi Road, Xi’an 710032, China

**Keywords:** lead neurotoxicity, ER stress, mitochondrial dysfunction, ubiquitination

## Abstract

Lead exposure may weaken the ability of learning and memory in the nervous system through mitochondrial paramorphia and dysfunction. However, the underlying mechanism has not been fully elucidated. In our works, with SD rats, primary culture of hippocampal neuron and PC12 cell line model were built up and behavioral tests were performed to determine the learning and memory insults; Western blot, immunological staining, and electron microscope were then conducted to determine endoplasmic reticulum stress and mitochondrial paramorphia and dysfunction. Co-immunoprecipitation were performed to investigate potential protein–protein interaction. The results show that lead exposure may cripple rats’ learning and memory capability by inducing endoplasmic reticulum stress and mitochondrial paramorphia and dysfunction. Furthermore, we clarify that enhanced MFN2 ubiquitination degradation mediated by PINK1 may account for mitochondrial paramorphia and endoplasmic reticulum stress. Our work may provide important clues for research on the mechanism of how Pb exposure leads to nervous system damage.

## 1. Introduction

Environmental and occupational lead contamination is harmful to human health. Many countries and regions have made a series of measures to control Pb contamination [1,2], but there are still many people are suffering from Pb contamination, especially in developing countries [3]. Pb exposure exerts irreversible damage effects on nervous system cognition, learning, and memory capabilities [4,5].

There is a consensus that Pb exposure damages children’s neurological and cognitive development, and can even lead to those children behaving antisocially and delinquently [6,7]. In recent years, there has been Pb contamination in drinking water in many U.S. cities, which has aroused public attention to the threat of Pb contamination to human health [8,9].

Mitochondria move dynamically in a cell so as to ensure their best performance. With the help of fusion and fission, mitochondria can ensure that they can move steadily. It has been revealed that steady movement is a precondition for mitochondria’s self-healing and performance of their function [10,11]. Mitofusin2 (MFN2) fusion protein plays an important role in adjusting mitochondria’s dynamic movement and maintaining their dynamic balance. The mitochondrion is an important organelle for neurons to grow and perform their function [12]. Mitochondria’s stable morphology is the basis for a neuron to accomplish physiological metabolism. Studies report that the pathology of many neurodegenerative diseases is related to mitochondrial dysfunction; thus, mitochondrial dysfunction has become a key target for curing neurodegenerative diseases [13].

Similar to mitochondria, the endoplasmic reticulum is also an important organelle. Interaction between endoplasmic reticulum and mitochondria exists in physiological processes. More and more evidences show that there is bridge between endoplasmic reticulum and mitochondria, which provides the precondition for their close interaction [14]. Studies report that as for Ca^2+^ storage in a cell, the endoplasmic reticulum supplies Ca^2+^ to mitochondria so that mitochondria can perform their function as usual [15]. Additionally, the endoplasmic reticulum is a significant site for proteins to be compounded, processed, and modified, so that the endoplasmic reticulum can provide proteins to mitochondria. In recent years, more and more studies have been conducted on the relationship between endoplasmic reticulum and mitochondria, which have confirmed that the endoplasmic reticulum regulates the dynamic movement and function of mitochondria [15], but the mechanism between them is unclear. In our previous research, it was found that Pb exposure results in endoplasmic reticulum stress. Meanwhile, Pb exposure has also been shown to cause mitochondrial dysfunction [16]. Therefore, whether lead-induced ER stress is closely related to mitochondrial dysfunction is worthy of further study.

Pb exposure results in the reduction of learning and memory capability in the nervous system, representing a breeding background for neurodegenerative diseases. However, the mechanism of influence on neuron damage from Pb exposure remains unclear. In this study, we established three types of models: an SD rat model, a primary culture model of hippocampal neurons, and a PC12 cell line model. Furthermore, we have focused on the condition of neuron damage and mitochondrial paramorphia and dysfunction after Pb exposure, so as to uncover whether endoplasmic reticulum stress induced by Pb exposure is the key factor for mitochondrial abnormality in dynamic fusion and fission. Furthermore, we illuminate that the ubiquitinational degradation of MFN2 is a key step leading to mitochondrial paramorphia and dysfunction. In conclusion, this paper provides important clues for research on the mechanism of how Pb exposure leads to nervous system damage.

## 2. Results

### 2.1. Pb Exposure Leads to Neurotoxicity and ER Stress

To investigate the neurotoxicity of Pb, we fed 21-day-old SD rats with increasing concentrations of PbAc (0, 100, 200 and 300 ppm) for 8 weeks. The result of atomic absorption spectrometry showed a significant increase in blood Pb and brain Pb concentrations (Figure 1A,B). The Morris water maze experiment also showed impaired spatial learning and memory ability of PbAc-treated mice (Figure 1C,D). Consistently, an escalating concentration of PbAc inhibited cell viability and led to cell damage in the primary culture of rat hippocampal neurons and in the PC12 cell line (Figure 1E–J). Morphological observations further indicated signs of apoptotic cell death, which has been confirmed by flow cytometry (Figure 1K).

PbAc treatment also resulted in morphological alterations in cellular ultrastructure. In comparison with control rats treated with a vehicle, an obvious change was the extension of endoplasmic reticulum (ER) lumen in the hippocampal region of Pb-exposed rats (Figure 2A). In agreement with this, in vitro exposure to Pb led to swelling of the ER in PC12 cells (Figure 2A). We therefore determined that Pb induced ER stress. Western blot analysis of rats’ hippocampus extract showed a significant increase in ER chaperone expression after 8 weeks of the PbAc regimen, which was accompanied by activation of C/EBP homologous protein (CHOP) signaling (Figure 2B). In vitro experiments with rat hippocampal neurons and PC12 cells also indicated the induction of ER stress (Figure 2C–F). Of note, ER stress occurred as early as 6 h of PbAc treatment, which may be an immediate response to Pb exposure that determines cell fate and mediates neurotoxicity.

### 2.2. Pb Exposure Disrupts Mitochondrial Morphology and Decreases MFN2 Protein Expression

Upon examination of ER morphology, we also noticed significant changes in mitochondrial ultrastructure. PbAc treatment broke down the mitochondria of rat hippocampal neurons (8 weeks) and PC12 cells (48 h), as judged by lipid deposition, vacuole formation, and mitochondrial swelling (Figure 3A,B). Quantitative measurement of mitochondrial length using mito-tracker indicated the shrinkage of mitochondria after 24 h of PbAc treatment (Figure 3C,D). Notably, mitochondria fission was not detectable within 12 h, suggesting that there are potential timely regulatory pathways that determine the morphology and function of mitochondria during Pb exposure.

Mitofusion (MFN) protein is a transmembrane GTPase anchored in the external mitochondrial membrane. Numerous studies have highlighted the prominent roles of MFN2 in maintaining mitochondria homeostasis and function, while the disruption of MFN2 has been shown to lead to mitochondrial breakdown and is implicated in various disorders, including neurodegenerative diseases [17]. To address whether MFN2 contributed to Pb-induced neurotoxicity and mitochondrial fission, we examined its expression at mRNA and protein levels in rat hippocampal neurons and PC12 cells, respectively. Western blot analysis showed that consecutive feeding with PbAc for 8 weeks dose-dependently decreased the MFN2 protein level in rats’ hippocampus (Figure 3E,F). In agreement with our previous observation, the MFN2 protein level did not change at 12 h of PbAc treatment, while it decreased sharply at 24 h and 48 h. Although there was also a reduction in the MFN2 counterpart, Mitofusin1 (MFN1) protein, it was interesting to notice that the magnitude of protein level reduction was much more evident for MFN2 (Figure 3E,F). This phenomenon might be a consequence of the predominant expression pattern of MFN2 protein in most tissues and cells. We therefore focused on MFN2 in the following experiments.

### 2.3. Pb exposure Leads to Enhanced ER–Mitochondria Interaction

Having established that Pb exposure leads to ER stress and mitochondrial fission, we proposed that there is enhanced ER–mitochondria interaction after PbAc treatment. In Figure 4, ER and mitochondria are labeled with ER-tracker (blue signal) and Mito-tracker (green signal); their localization and co-expression were detected under confocal laser microscope. Intriguingly, 5 μM PbAc treatment promoted mitochondria overlapping with ER, and the overlapping area further increased when PbAc concentration was reduplicated in PC12 cells (Figure 4A,B). Quantitative analysis showed an approximately 5-fold increase in the overlapping area after Pb exposure (Figure 4C). To corroborate these findings, we performed immunogold-labeled electron microscopy; the result showed reduced distance between ER and mitochondria (indicated by yellow arrows) (Figure 4D). Moreover, the colloidal gold particle-conjuncted MFN2 was predominantly expressed in the outer membrane of mitochondria (indicated by red arrows), whereas the amount of MFN2 dots significantly reduced in the presence of PbAc. Strikingly, PbAc treatment also led to the redistribution of MFN2 dots to ER (indicated by blue arrows), raising the possibility that MFN2 mediates the interaction between ER and mitochondria.

### 2.4. Induction of ER Stress Is Required for Mitochondrial Dysfunction

Given that ER stress is an immediate response to PbAc treatment, we proposed that enhanced ER–mitochondria interaction might be a stepwise biological event involving the induction of ER stress and alterations in mitochondrial morphology/function. To verify this hypothesis, we used an ER stress inhibitor 4-PBA at the final concentration of 5 mM. Thirty minutes of 4-PBA pre-treatment efficiently blocked PbAc-induced CHOP phosphorylation in PC12 cells. In agreement with this, the induction of ER stress key molecule Grp78 was also attenuated by 4-PBA (Figure 5A,B). Of note, 4-PBA single-agent had no obvious effect on mitochondrial morphology, whereas it prevented PbAc-induced mitochondrial fission (Figure 5E). Upon examination of MFN protein level, we observed that the restoration of MFN expression, particularly for MFN2, was associated with the blockage of ER stress (Figure 5C,D). Furthermore, the inhibition of ER stress and the preservation of mitochondria integrity ameliorated ROS production and apoptotic cell death in PC12 cells (Figure 5F,G). These results suggest that the induction of ER stress is crucial for mitochondrial disruption and Pb-induced cell death, and blockage of ER stress restores MFN expression and thereby reduces neurotoxicity.

### 2.5. Loss of MFN2 Primes the Cytotoxicity of Pb

Having established MFN2 as the core responder to Pb-induced mitochondrial fission and ER–mitochondria interaction, we next determined the significance of MFN2 in cell fate after Pb exposure. MFN2 expression was upregulated and downregulated in PC12 cells by transfection of pCMV3-MFN2 overexpressing vector and siRNAs, respectively. In agreement with our previous observations, knockdown of MFN2 expression readily reduced PC12 cell viability, and the addition of PbAc further suppressed cell proliferation (Figure 6A,B). Moreover, PbAc in combination with MFN2 siRNAs caused more profound cell damage, as judged by LDH leakage in the supernatant of PC12 cells (Figure 6C,D). In contrast, ectopic expression of MFN2-rendered PC12 cells grew freely in PbAc-containing medium and prevented LDH leakage (Figure 6A–D), indicating that MFN2 is a dominant factor for Pb^2+^ cytotoxicity.

### 2.6. PINK1 Underlies MFN2 Downregulation after Pb Exposure

Finally, we explored the molecular mechanism underlying MFN2 downregulation as a response to PbAc treatment. Upon examination of MFN2 mRNA level, we observed that MFN2 transcription in hippocampal neuron primary culture and PC12 cell line was not affected by the treatment, indicating that transcriptional machinery is dispensable for MFN2 protein reduction. To address whether protein destruction was responsible for MFN2 downregulation, we tested the possibility of autophagy and ubiquitination—two critical post-transcriptional regulatory mechanisms in eukaryotic cells—in MFN2 proteolysis. Although PbAc readily triggered autophagy in PC12 cells, confocal laser scanning showed that MFN2 protein was not co-localized with autophagic vacuoles, as judged by anti-LC3 labeling (Figure 7A–D). These findings suggested that autophagy may not be correlated with MFN2 destruction. Intriguingly, MFN2 protein was preserved by blockage of proteasome, indicating that ubiquitination of MFN2 may underlie its destruction. Indeed, PbAc treatment promoted MFN2 protein undergoing ubiquitination. 

Recent studies have shown that MFN2 is primarily ubiquitinated by the E3 ligase Parkin, and the association of MFN2 and Parkin is greatly enhanced in cells overexpressing PTEN-induced putative kinase 1 (PINK1) [18]. Of note, PINK1 is prominently expressed in the mitochondria and cytoplasm, and is involved in the maintenance of mitochondrial integrity and function, as well as in full activation of Parkin E3 ligase activity [19]. We therefore proposed PINK1 as a potent regulator of MFN2 stability. Immunoprecipitation assay revealed that PbAc dose-dependently increased MFN2 binding with PINK1 (Figure 7F), which was further supported by the co-localization analysis (Figure 7E). Moreover, knockdown of PINK1 antagonized mitochondria fission, ER stress, and cell damage (Figure 8A–E). Collectively, these results indicate PINK1 as a critical regulatory machinery upstream of MFN2, and suggest that PINK1 could be used as a target to prevent Pb-induced neurotoxicity.

## 3. Discussion

Environmental lead contamination seriously threatens human health. From the aspect of lead neurotoxicity, we found that Pb contamination harms hippocampal neurons in primary culture and PC12 cell viability, which consequently reduces the learning and memory capabilities of SD rats. Meanwhile, our study revealed that Pb exposure induces mitochondrial paramorphia and dysfunction and endoplasmic reticulum stress. Furthermore, this paper reveals the close relationship that exists between endoplasmic reticulum stress and the dynamic stability of mitochondria; endoplasmic reticulum stress even plays an important role in the neurotoxic effects of Pb. Additionally, this paper further highlights that MFN2 ubiquitinational degradation mediated by PINK1 is key during the process of mitochondrial fusion and fission controlled by the endoplasmic reticulum.

Lead exposure ruins the nervous system so that mice’s spatial, learning, and memory capabilities are affected [20]. This study verifies the neurotoxic effects of lead with the help of experiments in vivo and in vitro. Through testing with the Morris water maze, we found that compared with the matched group, the incubation period of the lead exposure group was significantly lengthened. Through examination with the dyeing test on immunological tissue, we found that the lead exposure group’s neurons in the hippocampus were reduced significantly. After the FCM test and MTT test, we also found that lead exposure will result in damage to PC12 cells and hippocampal neurons in primary culture.

Endoplasmic reticulum stress brings about neuron damage. Different kinds of nervous system diseases follow from endoplasmic reticulum stress [21]. This study also shows that lead exposure results in endoplasmic reticulum stress. After endoplasmic reticulum stress, neuron autophagy and apoptosis take place.

Abnormal protein accumulation and misfolding in the cytoplasm are the main reasons for causing endoplasmic reticulum stress. Meanwhile, lead exposure can give rise to abnormal accumulation of α-syn protein and others, which may be symmetrical to the reason for causing endoplasmic reticulum stress [22]. After lead exposure, we found that both the endoplasmic reticulum morphogenesis and the relative proteins of endoplasmic reticulum stress changed both in vitro and in vivo, indicating that lead exposure has an effect on endoplasmic reticulum stress.

However, the specific mechanism resulting in neuron damage from endoplasmic reticulum stress remains unclear. During the process of testing endoplasmic reticulum stress, we also observed the change process of mitochondrial morphogenesis. The mitochondrion is the principal source for a neuron’s survival to gain enough energy. Mitochondria are always dynamically moving, with both fusion and fission [23]. Therefore, when the dynamic movement of a mitochondrion is abnormal, mitochondrial dysfunction will take place [10]. An electron microscope can observe mitochondrial structure clearly. Through mito-tracker dyeing, the basic morphogenesis of mitochondria can be observed.

Comparing the mitochondrial morphogenesis and structure in the lead exposure group with that in the matched group, we found that mitochondria swell and vacuolization takes place. Furthermore, the average length of a mitochondrion is decreased. The main elements of controlling a mitochondrion’s dynamical stability include fusion and fission proteins. We used western blotting to test the changeable situation of mfn2 fusion protein in mitochondria.

Therefore, we speculate that the decrease in mfn2 protein is the main reason for causing mitochondrial dysfunction under the condition of lead exposure. Using an mfn2 overexpression plasmid vector, we structured PC12 cell with mfn2 overexpression and then applied the condition of lead exposure. Finally, we found that the toxic effects to PC12 cells by imposed by lead decreased in the mfn2 overexpression group. Both endoplasmic reticulum and mitochondria are important organelles, and their structures and functions interact with each other in the cytoplasm.

The endoplasmic reticulum can control mitochondrial morphogenesis and function by adjusting the calcium concentration in mitochondria [24]. It remains unclear whether mitochondrial paramorphia and dysfunction are related to endoplasmic reticulum stress. Using a transmission electron microscope, we observed the structure connection between endoplasmic reticulum and mitochondria. We found that the transactional distance between them is shortened. Additionally, after chemical dyeing on immunologic tissue, it was obvious that the transactional area is extended, which further discloses the close relationship between endoplasmic reticulum and mitochondria under the condition of lead exposure.

It has been shown that mfn2 fusion protein is closely related to the endoplasmic reticulum [25], but it is unclear whether endoplasmic reticulum can adjust mfn2. In order to discover the relationship between endoplasmic reticulum stress and mitochondrial dysfunction under the condition of lead exposure, we used 4-BPA inhibitor to treatPC12 cells and then expose cells to lead. After that, we found that the decrease in mfn2 was inhibited, and that mitochondrial paramorphia and dysfunction were controlled effectively within a short period of time, which illustrates that endoplasmic reticulum stress takes part in the process of mitochondrial paramorphia and dysfunction.

An IEM (immune electron microscope) is good for observing the molecule distribution specifically marked and illustrating spatial orientation. In this study, we used an IEM to identify the distribution and spatial orientation of mfn2 after lead exposure. Previous research has shown that the mfn2 molecule is a kind of fusion protein which always sticks to mitochondrial membranes [25].

However, in our study, we found that after lead exposure, mfn2 molecules not only stick to mitochondrial membranes, but are also present on the lumen of endoplasmic reticulum, which means that endoplasmic reticulum may take part in the process of the synthesis and degradation of mfn2. Therefore, we conclude that the decrease in mfn2 induced by lead exposure is related to protein degradation caused by endoplasmic reticulum stress.

There are two ways for protein degradation in cells. One is ubiquitinational degradation mediated by proteasome. The other is autophagy degradation mediated by lysosome. These two types of degradation mode are closely related to endoplasmic reticulum stress [26,27]. In our previous study, we found that lead exposure is bound to bring about the occurrence of endoplasmic reticulum stress and autophagy [22].

Therefore, we firstly analyzed whether the decrease in mfn2 protein is related to autophagy degradation. LC3II is the key marker to mark autophagosomes. Autophagosomes are the key to degrade abnormal proteins. We used the immunofluorescence histochemical staining method to dye mfn2 and LC3II. After that, we found that mfn2 and LC3II are not in colocation, which indicates that there may be no relationship between the increase in autophagy level induced by lead exposure and the degradation of mfn2 molecules.

There are three ways for autophagy degradation. One is that an autophagic vacuole integrates with a lysosome, which is called macroautophagy. The other is that an undegraded protein integrates with a lysosome, which is called microautophagy. The third one is that a molecular chaperone integrates with a lysosome after being mediated, which is called autophagy mediated by molecular chaperone [28]. In this paper, we focused on the relationship between macroautophagy and mfn2 molecules. Therefore, it cannot be completely confirmed that mfn2 degradation is realized without autophagy–lysosome interaction.

We observed mfn2’s ways of ubiquitinational degradation. A protein should complete its self-ubiquitination before ubiquitinational degradation. We used western blotting to test mfn2’s level of ubiquitination. The results show that mfn2’s level of ubiquitination in the lead exposure group is significantly higher than that in the matched group, which proves that mfn2 molecules in the lead exposure group may have been ubiquitinationally degraded. The precondition for protein to form ubiquitinational degradation needs C terminal of ubiquitin connecting with cysteine residue of no-specificity UAE (ubiquitin-activating enzyme), which can form E1 ubiquitin compound [29].

Then, E1 ubiquitin compound transfers ubiquitin to E2 ubiquitin-conjugating enzyme. E2 can transfer ubiquitin to the E-amine group of the target protein’s lysine residue. As usual, the target protein needs specific E3 ubiquitin protein ligase to realize ubiquitination [29]. No one has conducted any research on whether there exist some key proteins controlling mfn2 degradation induced by lead exposure.

It has been indicated in many studies that Parkin molecules are an important E3 ligase and take part in the process of ubiquitin degradation [30]. The function of PINK1 (PTEN-induced putative kinase 1) is similar to Parkin proteins, which are both recessive inheritance genes of autosomes related to Parkinson’s disease. PINK1 is widely expressed, especially in the heart, muscle, brain, and other such kinds of high-energy-consuming organs. PINK1 is usually distributed in inner mitochondrial membranes [31].

During the process of cell stress, PINK1 participates in mitochondrion autophagy so that it can provide protection for mitochondria [31]. In our study, we discovered that the expression level of PINK1 molecules in the lead exposure group was higher than that in the matched group. We used the co-immunoprecipitation method to detect the mutual relationship between PINK1 and mfn2, and the result shows that there exists a mutual relationship between them after lead exposure.

We applied siRNA to interfering PINK1 molecules and then exposed PINK1 molecules to lead. After that, we found that the decreased speed in mfn2’s expression level improved and that mfn2’s ubiquitin level decreased. The result also indicates that PINK1 participated in the mfn2 molecular decreasing process induced by lead exposure, and that the process is likely realized through ubiquitin degradation. Meanwhile, the experiment result indicates that PINK1 can adjust mfn2 protein’s degradation. Although this result was found under the condition of lead exposure, this degradation process may be part of an existing mechanism, revealed by simulation from lead exposure.

It has been shown that PINK1 molecules take part in mitochondrial autophagy [32]. However, PINK1 interacts with mfn2 without macroautophagy under the condition of lead exposure. Instead, PINK1 induces the presence of ubiquitin degradation through mediating the ubiquitin of mfn2 molecules. In conclusion, our study has explored the mechanism existing in the process of PINK1 adjusting mfn2, which fills the gap in this field and provides an important clue for research on how PINK1 controls the mitochondrial morphology and function.

It is time to analyze the dynamic change and important function of the interaction between endoplasmic reticulum and mitochondria under the condition of lead exposure. Meanwhile, this paper reveals that endoplasmic reticulum stress may be related to the chaos of mitochondrial fusion and fission and the decrease in mfn2 protein. Furthermore, this paper also demonstrates that the decrease in mfn2 protein mediated by PINK1 is key for the endoplasmic reticulum to control mitochondrial morphology and function. Therefore, this paper fills in several important research gaps and provides an important clue for research on the mechanism of lead-induced nerve toxicity and protection from it. Endoplasmic reticulum stress and PINK1 molecules may become important targets for preventing and curing lead-induced nerve toxicity.

## 4. Materials and Methods 

### 4.1. Experimental Animals and Treatments

The Institutional Animal Care and Use Committee of the Fourth Military Medical University (FMMU) approved all experiments involving animals. For our purposes, 21-day old SD rats were procured from the animal lab center of the Fourth Military Medical University and were used in the establishment of in vivo model. The rats were randomly divided into four groups: control, low-dose Pb exposure group, middle-dose Pb exposure group, and high-dose Pb exposure group. The number of experimental animals in each group was 12. Lead acetate was dissolved with deionized water to different concentrations (0, 100 ppm, 200 ppm, 300 ppm). The control group drank deionized water while using narcotics. The exposure lasted 8 weeks. The dosage of Pb was based on previous studies published in 2012 [20].

### 4.2. Quantification of Behavioral Tests

The Morris water maze behavioral test was performed as described previously [33]. The rats were placed in a pool of water containing a platform that is held in a constant position just below the surface of the water during the acquisition trials. An above-water video camera monitored the mice from a platform with a diameter of 10 cm. The trial involved placing a mouse in the water at one starting point. Animals that had difficulty finding the platform within two minutes were guided to it, where they stayed for 10 s. A 120 s swim was allowed after the hidden platform test, without any restrictions on the mice. Swim speed, escape latency, and distance traveled were measured during each trial. Each quadrant’s entries, swimming speed, and time were recorded with cameras. Generally, the rats were able to determine the location of the hidden platform from distal visual cues arranged around the room. The rats escaped from the maze when they found the platform. In order to analyze these data, we used Noldus Ethovision XT (Wageningen, The Netherlands).

### 4.3. Lead Concentration Analysis

Blood samples (600 *l/mouse) for measuring blood lead levels were collected from the left ventricle. Lead was extracted from brain tissues by drying them and digesting them with organic solubilizers. Lead levels were measured in duplicate using a graphic furnace and an atomic absorption spectrometer (AAS) (PerkinElmer 800, Norwalk, CT, USA).

### 4.4. Cell Culture and Treatments

As in in vitro models, a PC12 cell line was purchased from the Institute of Basic Medical Sciences of the China Science Academy. PC12 cells were cultured in high-glucose Dulbecco’s modified Eagle’s medium (DMEM) supplemented with 10% fetal bovine serum (FBS). The final concentration of PbAC2 was made by dissolving it in distilled water at a concentration of 1 mM, and then diluting it with serum-free DMEM. In our previous study, lead acetate (20 μM) and a 24 h stimulation period were used for lead stimulation.

4-PBA (sodium phenylbutyrate) was procured from Sigma-Aldrich (St. Louis, MO, USA) and dissolved with 0.9% NaCl. 4-PBA (5 mM) was added for 4 h before the treatment of Pb. In our preliminary experiments, we applied the 4-PBA concentration according to the manual.

Sprague–Dawley rats were purchased from BioLASCO Taiwan Co., Ltd. Primary cortical and hippocampal neurons were dissociated from dissected cortex and hippocampus, respectively, from rat embryos (embryonic day 18, E18), as previously described [34,35]. Cells were seeded on poly-l-lysine coated plates. On in vitro day 0 (DIV0), primary neurons were cultured in MEM supplemented with 5% FBS, 5% HS, and 100 units/mL PS under 5% CO_2_ conditions. The culture medium was changed to Neurobasal medium containing 25 μM glutamate, 2% B27, 0.5 mM l-Gln, and 50 units/mL PS on DIV1. AraC (10 μM) was added to neurons on DIV2 to inhibit proliferation of glial cells. On DIV3, the medium was changed to Neurobasal medium containing 2% B27, 0.5 mM l-Gln, and 50 units/mL PS. After DIV3, the medium was half-changed with Neurobasal/Glutamine medium every 3 days. 

### 4.5. Measurement of Cell Viability by MTT Assay

The PC12 cells (control/untransfected cells/NC/siRNA) (2.0 × 10^3^ cells/well) were cultured in DMEM supplemented with 10% fetal bovine serum (FBS; Gibco; Thermo Fisher Scientific, Inc., Waltham, MA, USA) in 96-well plates (Costar; Corning, Inc., Corning, NY, USA). In addition to the 15-day incubation period, 10 * l of MTT (Sigma-Aldrich; Merck KGaA, St. Louis, MO, USA) solution was added to the wells at a concentration of 5 mg/mL in ddH_2_O. During the incubation period of four hours, the plates were incubated at 37 °C. To each well, 100 μL of DMSO was added to dissolve the intracellular formazan crystals. Cell proliferation was evaluated on a microplate reader (BioTek Instruments, Inc., Winooski, VT, USA) set to 490 nm [36].

### 4.6. Western Blotting

After each treatment, western blotting was performed as previously described [37]. We extracted total proteins from the hippocampus tissues and the PC12 cells by lysing them on ice in a lysis buffer containing protease inhibitors following treatment. Measurement of protein concentration was conducted using a Pierce BCA Protein Assay Kit (Thermo Fisher Scientific, Waltham, MA, USA). Separation was performed by sodium dodecyl sulfate + polyacrylamide gel electrophoresis, followed by transfer to PVDF membranes. The membrane was blocked in 5% no-fat milk in TBS buffer incubated with Grp78 (Abcam, ab21685), Beclin-1 (CST, #3738), LC3 (CST, #3868), P-PERK (CST, #3179S), PERK (CST, #31925), PINK1 (Abcam, ab137361), P-CHOP (Sigma-Aldrich, SAB4301473), MFN2 (Abcam, ab124773), MFN1 (Abcam, ab 104274), and Ubquitin (Abcam, ab140601); β-actin (CST, #8242) was selected as an internal control. The PVDF membranes were stripped if target proteins weights were close to each other. The membranes were soaked in western blot stripping buffer (Thermo Scientific, Waltham, MA, USA) for 15 min, rinsed with water, and then dried. An enhanced chemiluminescence system (Bio-rad, Hercules, CA, USA) was used to visualize the protein bands, and ImageJ was used to calculate the relative protein density.

### 4.7. Measurement of Apoptosis by Flow Cytometry

Measurement of apoptosis was performed with a FITC annexin V–fluorescein isothiocyanate (FITC) kit (Key GEN Bio TECH, Nanjing, China). Flow cytometry analysis of PC12 cells was performed in the four treatment groups at 3, 6, and 9 h post exposure, followed by two washes with pre-chilled PBS and staining with FITC-conjugated annexin V and propidium iodide. Within an hour, samples were analyzed using a BD Biosciences FACScalibur flow cytometer equipped with Flow Jo software (v10, BD Biosciences, NJ, USA).

### 4.8. Immunofluorescence Staining

To fix cells, paraformaldehyde (4%) was used at room temperature for 20 min. Non-specific antibodies were blocked for 20 min with PBS containing 0.1% Triton-X 100 and 1% FBS. The primary antibodies (Rabbit anti-Mfn2 antibody from Abcam, UK; mouse anti-PINK1 antibodies from Abcam, UK; Mito-Tracker from Invitrogen, UK; ER-Tracker from YEASEN, China) were incubated on the cells overnight at 4 °C. The secondary antibodies (Alexa 488 (green)-conjugated anti-rabbit IgG and Alexa 594 (red)-conjugated anti-mouse IgG antibodies (1:1000)) was incubated on the cells for 1 h at room temperature. Negative control groups were incubated with PBS in place of primary antibodies. The cells were analyzed and photographed under a ZEISS Axiovert 200 fluorescence microscope after the nuclei were counterstained with DAPI. A blank was used in all tests as a negative control, which was the same except that the primary antibodies were replaced with PBS. Images were analyzed using the Nikon’s universal software platform [38]. Nikon’s universal software platform, NIS-Elements, combines powerful image acquisition, analysis, visualization, and data sharing tools. With fully customizable user interfaces and software modules, NIS-Elements can serve as a simple interface for photo-documentation, power complexes, and conditional workflows, with automated imaging and analysis routines. Javascripting and graphical programming tools enable users to easily create custom analysis and acquisition workflows. 

### 4.9. Quantitative Real Time Polymerase Chain Reaction (qRT-PCR)

Isolation of cellular mRNA was performed using the Trizol reagent (Invitrogen, USA) after rinsing the hippocampus tissues or PC12 cells three times with PBS. A One Step Prime Script^®^ RNA cDNA Synthesis Kit (TAKARA, Kusatsu, Japan) was used to transcribe total RNA to cDNA, and the Nano Drop ND-100 Spectrophotometer (Nano Drop Technologies, Wilmington, NJ, USA) was used to quantify the amount of cDNA. In order to perform qRT-PCR, we used the SYBR Premix Ex Taq TM Kit (TAKARA) and the Applied Biosystems 7500 Fast Real-time PCR System (Applied Biosystems, Madrid, Spain). Primers were sequenced from the Primer Bank. PCR was conducted in strips, including a no-template control for each amplification. The samples were analyzed in triplicate. Melt-curve analysis was performed immediately following the amplification procedure to verify the specificity of the amplification [39].

### 4.10. RNA Interference

MFN2 small interfering RNA (siRNA), (Dharmacon RNA Technologies, L-012961-00). PINK1 siRNA ( 5′ -GCU GCA AUG CCG CUG UGU ATT-3′ ), and Negative Control (NC) siRNA (Dharmacon RNA Technologies, D-001810-10-05) were designed by and purchased from Gene Pharma (Gene Pharma, Shanghai, China). The siRNAs were dissolved in DEPC-treated water. Transfecting sub-confluent proliferating cells with Lipofectamine 2000 (Invitrogen, Carlsbad, CA, USA) was performed as directed by the manufacturer [40]. The siRNA was finalized at a concentration of 50 nM. Lentivirus-overexpressing MFN2 particles (Lenti-OE), overexpressing control particles (ctrl-OE), were purchased from Genechem (Shanghai, China). For infections, chondrocytes were incubated with lentiviral particles and Polybrene (1 mg/mL) in growth medium. After 12 h, the infection medium was replaced by growth medium.

### 4.11. Transmission Electron Microscopy (TEM) 

Cells from the hippocampal tissues and the PC12 cells were harvested and fixed in glutaraldehyde for 24 h at room temperature. The fixed cells were post-fixed in 1% osmium tetroxide, dehydrated using a graded ethanol series (30, 50, 70, 80, 90, and 100%), embedded in Epon (Energy Beam Sciences, Agawam, MA, USA), and sliced into ultrathin (50–60 nm) sections using a Leica EM UC6 ultramicrotome (Leica Microsystems, Wetzlar, Germany). We stained the sections with uranyl acetate and lead citrate and observed them with a transmission electron microscope (H7500; Hitachi, Tokyo, Japan) [41].

### 4.12. Immunoelectron Microscopy (IEM)

Fixing PC12 cells overnight in phosphate buffer (PH = 7.2) of 4% paraformaldehyde and 1% glutaraldehyde, followed by gradual ethanol dehydration, was performed. Embedded cells were polymerized under UVA at 20 °C in Lowicrys K4M resin. The cells were thinly sectioned to a thickness of 50–70 nm and incubated with a primary antibody (Mouse anti-Mfn2 from Abcam, USA) and secondary antibody (6nm Colloidal Gold AffiniPure Goat Anti-Mouse IgG (H+L) from Jackson Immuno Research, USA) on a nickel net. Images were observed and collected using a transmission electron microscope (Hitachi H-7500, HITACHI, Tokyo, Japan) after conventional staining [42].

### 4.13. Immunoprecipitation (IP)

PC12 cells were lysed with a buffer composed of 50 mM Tris−HCl, 150 mM NaCl, 1% Triton X, 2 mM EGTA, 1 mM MgCl_2_ with an addition of 10% glycerol, PIC (100×), 10 mM NEM, and 50 μM MG132. Protein extract (250 μg^−1^ mg) was incubated with 10 μL of protein A agarose beads (Roche), previously balanced in lysis buffer, for 30 min at 4 °C on the wheel (pre-cleaning). In parallel, 30 μL of balanced beads were incubated with 1 μg of PINK1 (Abcam, ab137361) or without the antibody (negative control) for 1 h and half at 4 °C on the wheel in 70 μL of lysis buffer. Pre-cleaning beads were centrifugated at 4000 rpm for 5 min, and the supernatant was incubated overnight at 4 °C with antibody-conjugated beads. The following day, the mixture was centrifuged and the supernatant was discarded. The beads were washed three times for ten minutes at 4 °C with lysis buffer and boiled for 10 min at 95 °C in 30–50 μL Laemmli loading buffer 2X (Laemmli 4X: Tris HCL 300 μM, pH 6.8; SDS 300 μM; Sucrose 1,4 M; Beta mercaptoethanol 8%; Bromophenol blue). After maximum speed centrifugation, the supernatant was collected and analyzed by western blotting on NuPAGETM 3–8% Tris-Acetate Protein Gels [43].

### 4.14. LDH Assay Method

LDH activity was measured using the Pierce LDH Cytotoxicity Assay Kit following the manufacturer protocol (ThermoFisher Scientific, Waltham, MA, USA). Briefly, 50 µL of conditioned medium (CM) was collected from each sample. The cells were lysed using 10 µL of the lysate buffer provided in the assay kit for 45 min at 37 °C, and 50 µL of lysed cells were collected from each sample. The LDH assay was run on the CM and lysate samples. Absorbance was measured at 490 and 680 nm on a SpectraMax iD5 (Molecular Devices, San Jose, CA, USA).

### 4.15. Measurement of Reactive Oxygen Species Generation

2′,7′-dichlorodihydrofluorescein diacetate (DCFH-DA) was used to detect intracellular ROS level in PC12 cells. Briefly, PC12 cells (10^6^ cells/mL) were incubated with 25 μM DCFH-DA in DMEM without FBS at 37 °C for 30 min, and then washed thrice with PBS. Cells were collected and the 2′,7′-dichlorofluorescein (DCF) fluorescence was observed with a microplate reader (Bio-Rad, USA) at an excitation wavelength of 488 nm and an emission wavelength of 525 nm.

### 4.16. Statistical Analysis

A minimum of three experiments were performed, the results of which are expressed as mean ± SEM. Real-time PCR data are shown using the 2^−ΔΔCt^ method. Post hoc analysis was conducted using a two-way ANOVA, followed by the Tukey test. Differences with p-values < 0.05 were considered statistically significant. All analyses were performed using the Statistical Package for the Social Sciences (SPSS) 20.0 software (SPSS Inc., Chicago, IL, USA). The specific indices of statistical significance are indicated in individual figure legends.

## Figures and Tables

**Figure 1 ijms-23-11435-f001:**
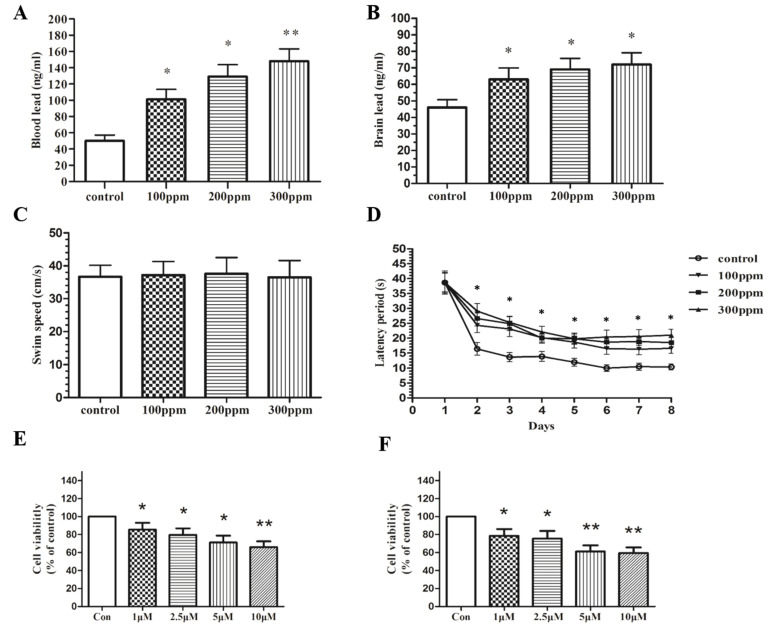
Pb exposure leads to neurotoxicity. Rats were given different doses of lead acetate in water for 8 weeks. After Pb exposure, levels of Pb in the blood (**A**) and brain (**B**) were determined by an atomic absorption spectrophotometer. The Morris water maze was used to measure the effect of Pb exposure on spatial learning and memory capacity in rats (**C**,**D**). Lead acetate (0, 1 uM, 2.5 uM, 5 uM, and 10 uM) was dissolved in DMEM and Neuron Basal culture medium. Primary hippocampal neurons were exposed to different concentrations of lead acetate for 12 h and 24 h. The effect of Pb on cell viability was determined by MTT (**E**,**F**). PC12 cells were exposed to different concentrations of lead acetate for 12 h and 24 h. The effect of Pb on cell viability was determined by MTT (**G**,**H**) and LDH (**I**,**J**) assay, and apoptosis was evaluated using flow cytometry (**K**). The data are expressed as mean ± SD of three independent experimtents. * *p* < 0.05, ** *p* < 0.01 and *** *p* < 0.001 compared with control group.

**Figure 2 ijms-23-11435-f002:**
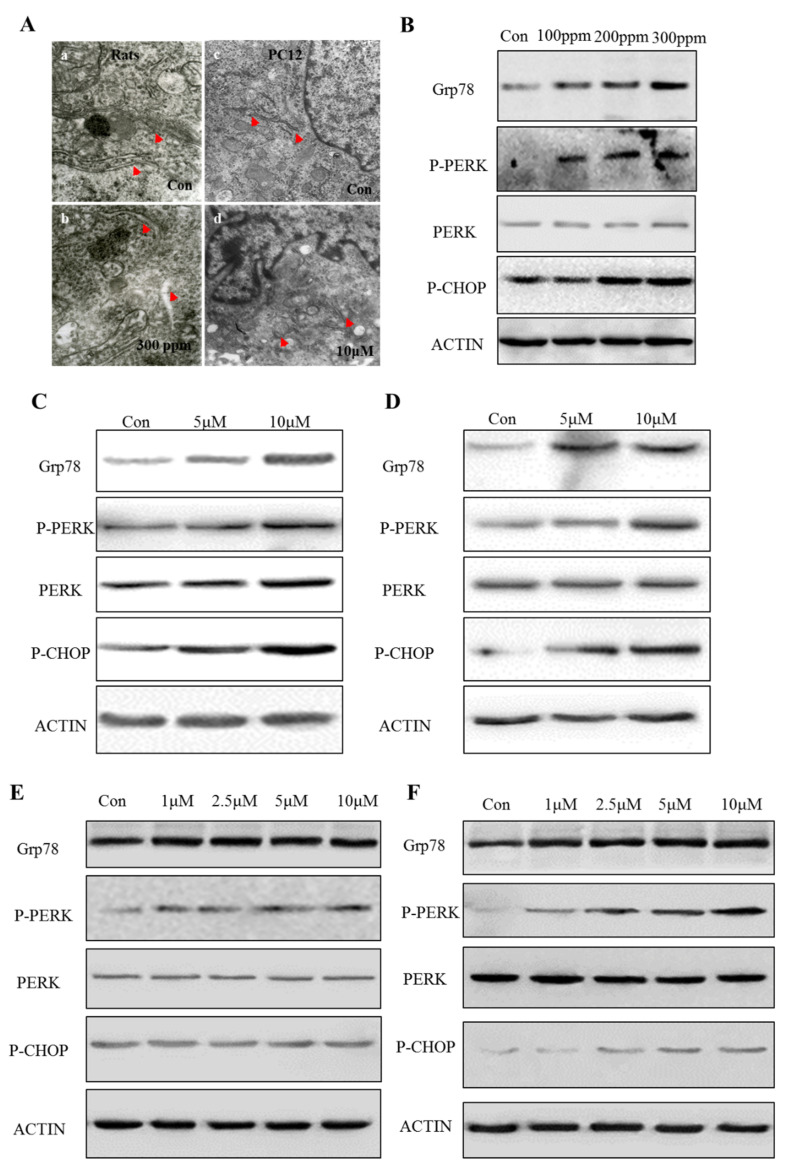
Pb exposure leads to ER stress. After Pb exposure, the changes in morphology of ER in rats and PC12 cells were observed under a high-resolution electron microscope (**Aa**–**Ad**). Western blotting was conducted to examine ER stress-related proteins and signaling pathways in SD rats (**B**). PC12 cells were treated for 12 h and 24 h with graded concentrations of Pb, and western blotting was conducted to examine ER stress-related proteins and signaling pathways in hippocampal neurons (**C**,**D**) and PC12 cells (**E**,**F**).

**Figure 3 ijms-23-11435-f003:**
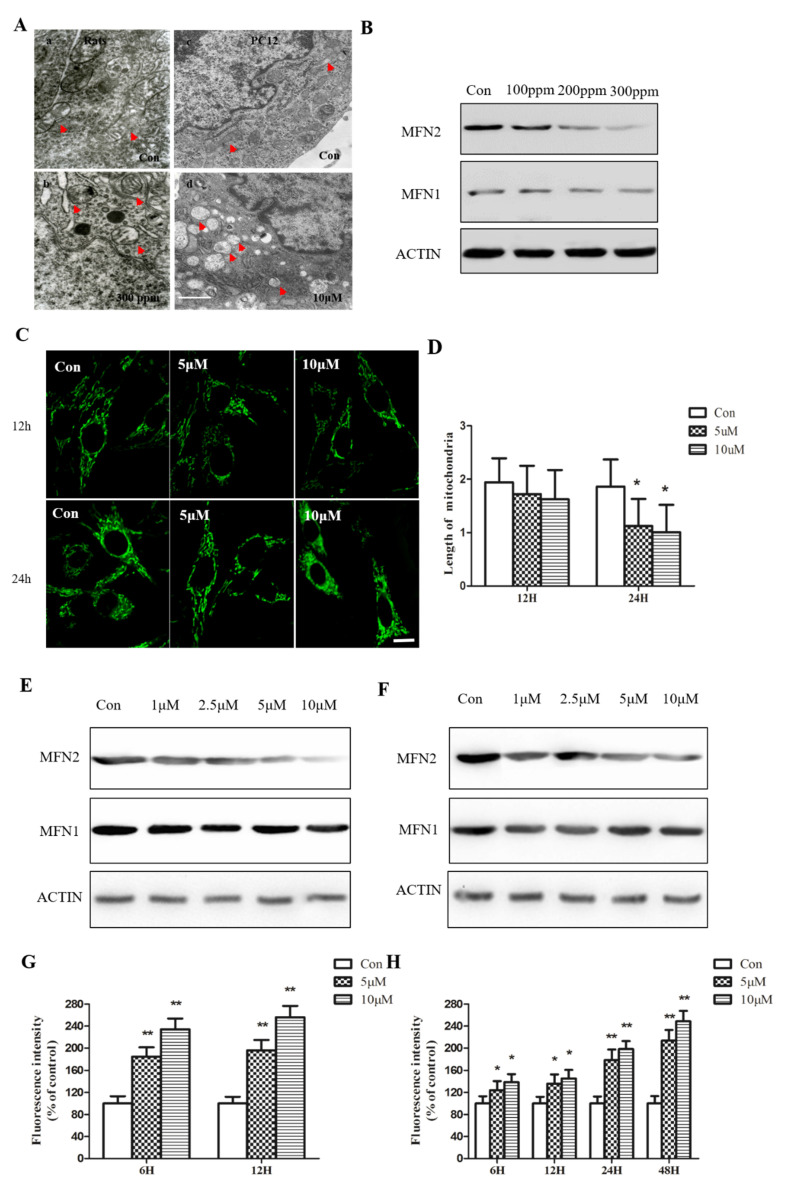
Pb exposure disrupts mitochondrial morphology and decreases MFN2 protein expression. After Pb exposure, mitochondrial morphology in rats and PC12 cells was observed under a high-resolution electron microscope (**A**). Western blotting was performed to assess the expression of MFN2 and MFN1 in rats (**B**). PC12 cells were treated for 12 h and 24 h with graded concentrations of Pb. Mitochondrial morphology in PC12 cells was observed using mito-tracker, which indicated the shrinkage of mitochondria after Pb exposure (**C**). Quantitative measurement of mitochondrial length by NIS-Elements (**D**). Western blotting was performed to assess the expression of MFN2 and MFN1 (**E**,**F**). ROS generation was measured by DCFH-DA probe, respectively (**G**,**H**). * *p* < 0.05 and ** *p* < 0.01 compared with control group.

**Figure 4 ijms-23-11435-f004:**
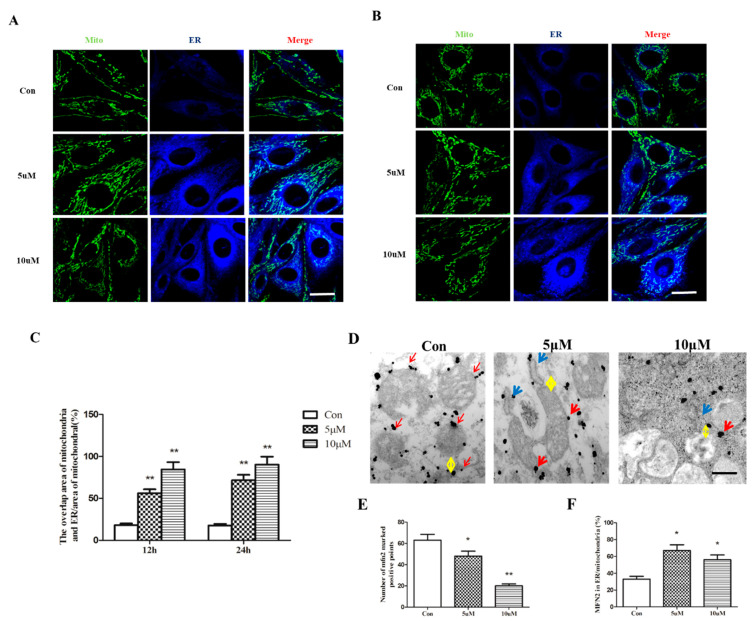
Pb exposure leads to enhanced ER–mitochondria interaction. PC12 cells were treated for 12 h and 24 h with graded concentrations of Pb. ER and mitochondria were labeled with ER-tracker (blue signal) and Mito-tracker (green signal); their localization and co-expression were detected under confocal laser microscope (**A**,**B**). The overlapping area of ER and mitochondria was quantitatively analyzed after Pb exposure (**C**). Distance between ER (indicated by blue arrows) and mitochondria (indicated by yellow arrows) was detected by Immunogold-labeled electron microscopy (**D**). Moreover, the distribution of MFN2 (indicated by red arrows) in cells was detected by colloidal gold particle-conjuncted MFN2 dyeing (**D**). The number of MFN2-marked positive points and MFN2 in ER/mitochondria were quantitatively analyzed after Pb^2+^ exposure (**E**,**F**). * *p* < 0.05 and ** *p* < 0.01 compared with control group.

**Figure 5 ijms-23-11435-f005:**
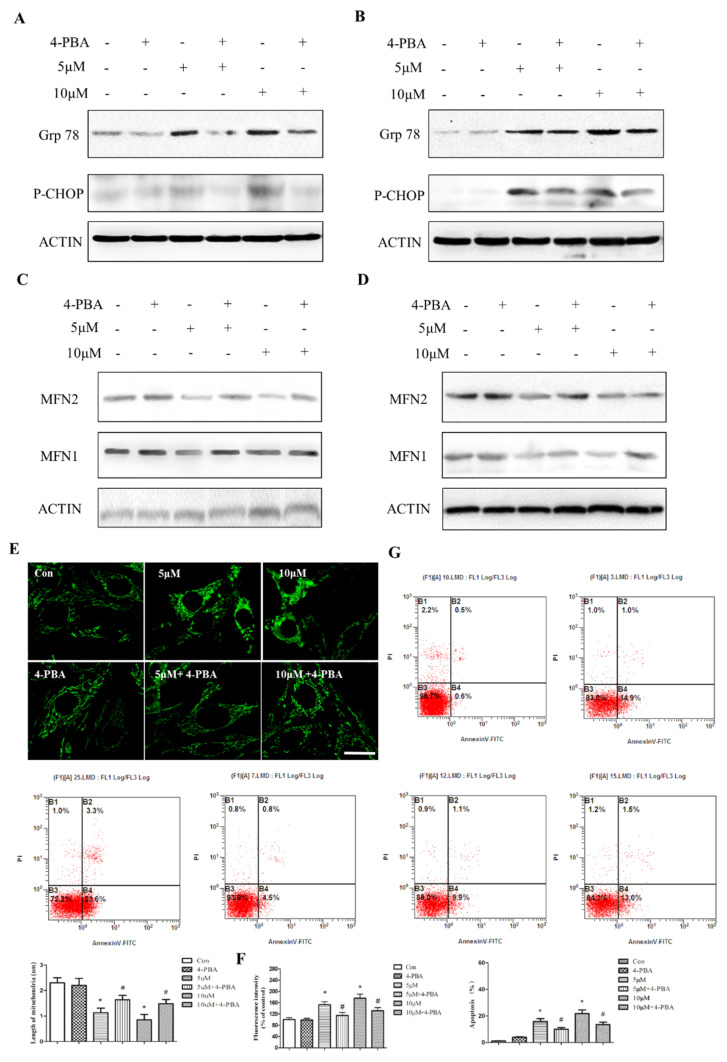
Induction of ER stress is required for mitochondrial dysfunction. PC12 cells were pretreated with 4-PBA (5 mM), an inhibitor of ER stress, and then exposed to Pb. Protein expression of ER stress-related proteins and MFNs were examined by western blotting (**A**–**D**). Mitochondrial morphology in PC12 cells was observed using mito-tracker, which indicated the shrinkage of mitochondria after Pb exposure. Quantitative measurement of mitochondrial length by NIS-Elements (**E**). ROS generation was measured by DCFH-DA probe (**F**). The effect of Pb on cell apoptosis was evaluated using flow cytometry (**G**). * *p* < 0.05 compared with control group; ^#^ *p* < 0.05 compared with Pb-treating group.

**Figure 6 ijms-23-11435-f006:**
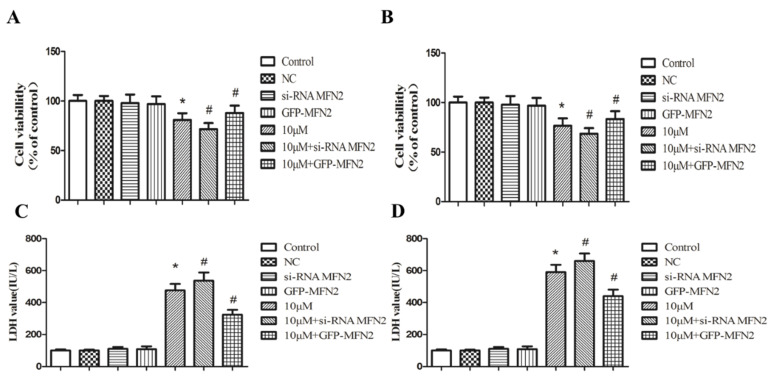
Loss of MFN2 primes the cytotoxicity of Pb. MFN2 expression was upregulated and downregulated in PC12 cells by transfection of pCMV3-MFN2 overexpressing vector and siRNAs, respectively. The effect of Pb on cell viability was determined by MTT (**A**,**B**) and LDH (**C**,**D**) assay. The data are expressed as mean ± SD of three independent experiments. * *p* < 0.05 compared with control group; ^#^ *p* < 0.05 compared with Pb-treated group.

**Figure 7 ijms-23-11435-f007:**
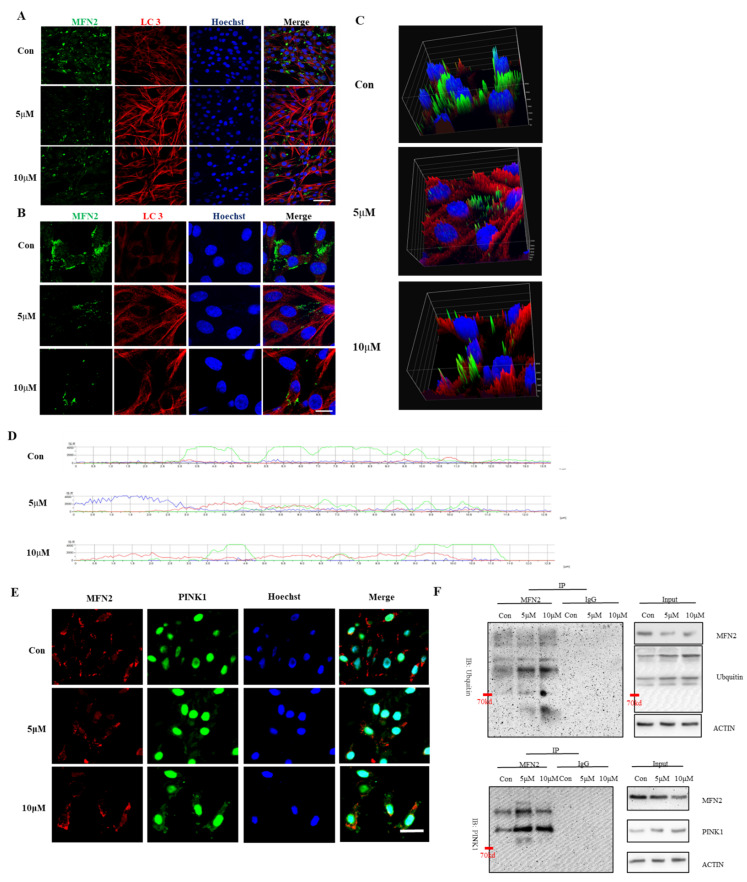
PbAc treatment promoted MFN2 protein undergoing ubiquitination. Western blotting and immunofluorescence were used to detect changes in autophagy degradation and the ubiquitin-proteasome pathway. Immunofluorescence was used to detect the co-expression of autophagy-related protein LC3 with mfn2 after Pb exposure for 12 h (**A**) and 24 h (**B**). Three-dimensional reconstruction of PC12 cells exposed to Pb for 24 h was performed using NIS-Elements (**C**), and the fluorescence co-expression of LC3 and Mfn2 proteins was analyzed by NIS-Elements (**D**). Immunofluorescence (**E**) and immunoprecipitation (IP) (**F**) were used to detect the co-expression of ubiquitin E3 ligase PINK1 with mfn2.

**Figure 8 ijms-23-11435-f008:**
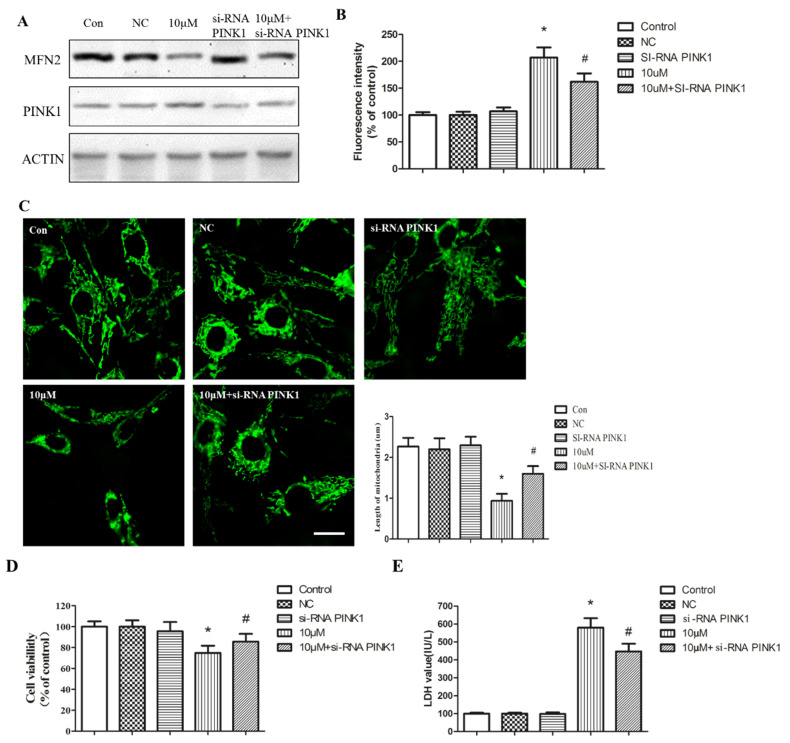
PINK1 underlies MFN2 downregulation after Pb exposure. After interference of PINK1, western blotting was performed to assess the expression of MFN2 (**A**). ROS generation was measured by DCFH-DA probe (**B**). Mitochondrial morphology in PC12 cells was observed using mito-tracker, which indicated the shrinkage of mitochondria after interference of PINK1 and Pb exposure. Quantitative measurement of mitochondrial length by NIS-Elements (**C**). The effect of Pb on cell viability was determined by MTT (**D**) and LDH (**E**) assay. The data are expressed as mean ± SD of three independent experiments. * *p* < 0.05 compared with control group; ^#^ *p* < 0.05 compared with Pb-treatment group.

## Data Availability

The data that support the findings of this study are available from the corresponding author upon reasonable request.

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
