# Peer review of "Lead Disrupts Mitochondrial Morphology and Function through Induction of ER Stress in Model of Neurotoxicity"

_ijms, 2022, doi:10.3390/ijms231911435_

Round 1

Reviewer 1 Report

The study by Zhang et al., titled as “Lead disrupts mitochondrial morphology and function through induction of ER stress in the model of neurotoxicity” contains interesting perspectives of how Pb exposure can lead to nervous system damage via disruption of the mitochondrial morphology and function. However, I have few concerns which are listed below for authors to improve this manuscript –

Major Concerns-

1-    Authors are suggested to define all the abbreviations when they are first introduced/used.

2-    In the materials and methods section authors are suggested to make changes as follow-

a.    How many SD rats were included in each group?

b.    Include LDH assay method briefly.

c.    Explain hippocampal neuron isolation method and culture conditions.

d.    Explain the Morris water maze experimental conditions.

e.    Explain the ROS methodology.

Minor Concerns-

1-    There are lots of grammatical errors and unclear sentences in the introduction. Authors are suggested to re-write it using small sentences. Few corrections are suggested below-

a.    Line 35- fuse should be replaced with fusion.

b.    The sentence written in the line 50-52 is not clear, need to be restructured.

c.    The sentence written in the line 54-56 is not clear, this also need to be restructured.

d.    In the line 75 “leading” can be changed to “leads”

2-    Figure 4A has duplicate image. Authors are suggested to change it.

3-    Authors are suggested rearrange the figure 7E same as 7A.

4-    Figure 3B B: has no panel description in the legends. Authors are suggested to explain it.

5-    Figure 3D is too small. It can be enlarged to make it clearer. May be best at least to match the sizes of other panels/ figures.

Author Response

We thank you and the reviewers for your constructive review of our work, as well as the opportunity to submit a revised manuscript. Your suggestions are extremely valuable and greatly appreciated. We have made every effort to address your comments, which has been summarized here and incorporated/highlighted in red in the manuscript.

Reviewer 2 Report

The general concern for this submission is related to description of methods and alignment of the Results narrative with the specific figures the authors are discussing. Throughout the manuscript, the authors do not direct the reader to the specific figures that correlate with their results. Related to this, there is minimal or missing information regarding dose, time, as well as other important elements of experimental description that is missing from the Results and Figure Legends. This needs to be resolved to help the reader understand the results.

-Within the Introduction, there is a paragraph discussion about ubiquitin degradation. However, this is not addressed in any of the experiments.

-Also in the Introduction, there is discussion of neurodegeneration, which is also out of place as they are researching lead, which is more often associated with learning and memory issues during neurodevelopment.

-The description of the flow cytometry appears to be from a different study as it references treatment with E. coli.

-The authors should pick either 12hrs or 24hrs of treatment and only report these findings. Currently, 12hrs and 24hrs are reported and it is too much and does not add to the story.

-Figure 1D: Are the differences statistically significant between the control and treated groups? No indication is given.

-The methods for isolation and creation of the hippocampal primary cultures is not provided.

-Since this paper is so focused on ER stress, more ER stress-related markers need to be evaluated. For example, IRE1, ATF6, ATF4, IEF, XBP1, in order to provide a more comprehensive understanding of the ER stress that is claimed to occur.

-The authors rely heavily on the PC12 cell line. However, the alterations to GRP78 are not very robust in this cell line, no matter the concentration of lead or the time of treatment. This is concerning as ER stress and this cell line are used primarily to illustrate the role of lead in ER stress.

-What is NIS Element and how does it assess mitochondrial length? The changes to MFN2 at 12 and 24hrs are pretty profound. How does MFN2 expression relate to mitochondrial length?

-Figure 4: It appears that the ER tracker completely illuminates the entire cell, meaning it is not a useful marker for ER stress, in the context of assessing the overlap of mitochondria on ER.

-The EM images in Figure 4 are minimally informative. How were any of these measurements performed? How many sections were used? How many regions of interest? Without this information, these images are not very helpful.

-Similar concerns about NIS Elements in Figure 5.

-Figure 6: Images need top be shown that demonstrate the authors' ability to knockdown MFN2 and overexpress MFN2 in PC12 cells.

-

Author Response

(The authors gave the same response as above.)

Round 2

Reviewer 1 Report

No concerns.

Reviewer 2 Report

The authors have addressed all concerns.